# Identification and characterization of a target antigen recognized by the monoclonal antibody against *Opisthorchis viverrini*

Anchalee Techasen[1,2], Chanika Worasith[1,3], Duangkamon Muengsaen[1], Jiraprapa Ponglong[1], Panupong Mahalapbutr[4], Napat Kongtaworn[5], Thanyada Rungrotmongkol[5,6], Kanoknan Khongsukwiwat[1,7], Phattharaphon Wongphutorn[1,7], Chompunoot Wangboon[8], Chutima Homwonk[1,7], Sujittra Chaiyadet[9], Thewarach Laha[7], Sutas Suttiprapa[9], Chadamas Sakonsinsiri[1,4], Paiboon Sithithaworn[1,7*◉], Raynoo Thanan[1,4*◉]

1 Cholangiocarcinoma Research Institute, Khon Kaen University, Khon Kaen, Thailand, 2 Faculty of Associated Medical Sciences, Khon Kaen University, Khon Kaen, Thailand, 3 Department of Adult Nursing, Faculty of Nursing, Khon Kaen University, Khon Kaen, Thailand, 4 Department of Biochemistry, Faculty of Medicine, Khon Kaen University, Khon Kaen, Thailand, 5 Program in Bioinformatics and Computational Biology, Graduate School, Chulalongkorn University, Bangkok, Thailand, 6 Center of Excellence in Structural and Computational Biology, Department of Biochemistry, Faculty of Science, Chulalongkorn University, Bangkok, Thailand, 7 Department of Parasitology, Faculty of Medicine, Khon Kaen University, Khon Kaen, Thailand, 8 School of Preclinic, Institute of Science, Suranaree University of Technology, Nakhon Ratchasima, Thailand, 9 Department of Tropical Medicine, Faculty of Medicine, Khon Kaen University, Khon Kaen, Thailand

◉ These authors contributed equally to this work.
* raynoo@kku.ac.th (RT); paib_sit@kku.ac.th (PS)

## Abstract

*Opisthorchis viverrini* (Ov) infection caused opisthorchiasis, which posed an important risk for the development of cholangiocarcinoma (CCA). Therefore, it is crucial to focus on the primary prevention and control of opisthorchiasis in order to control CCA effectively in Thailand and other endemic regions. A recent diagnostic method of antigen detection using monoclonal antibody-based enzyme-linked immunosorbent assay (mAb-ELISA) has the potential for rapid mass screening of opisthorchiasis. Nevertheless, the specific antigen(s) in Ov adult worms recognized by mAb have not been determined. In this study, we aimed to identify and characterize the target molecule of our in-house Ov-specific monoclonal antibody (mAb KKU505). The specific antigenic band formed by the reaction of Ov adult worm extract and mAb KKU505 was detected using western blot analysis. The protein band was identified as the myosin heavy chain of Ov using LC-MS/MS analysis. The reactivity of the recombinant full-length myosin heavy chain (rMHC) was comparable to that of the crude Ov antigen when evaluated using mAb-ELISA at similar protein concentrations. Moreover, the binding ability between Ov myosin head domain and mAb KKU505 was confirmed using *in silico* analysis. The results reported here indicate that rMHC could potentially substitute for Ov crude antigen in antigen detection by mAb-ELISA

**Data availability statement:** "All relevant data are within the paper and its Supporting Information files".

**Funding:** This research was supported by the National Research Council of Thailand (NRCT) to PS and AT via Cholangiocarcinoma Research Institute, Khon Kaen University and the Fundamental Fund of Khon Kaen University from the National Science to RT, Research and Innovation Fund (NSRF) to PS.

**Competing interests:** The authors have declared that no competing interests exist.

and as a positive control for Ov-strip in lateral flow assays, thereby avoiding the use of laboratory animals for the production of Ov adult worms.

## Introduction

*Opisthorchis viverrini* (Ov) infection is a major factor contributing to the development of cholangiocarcinoma (CCA) in the Greater Mekong sub-region, especially north-eastern Thailand [1]. Raw cyprinoid fish consumption is a high risk for Ov infection. Up to date, many studies revealed that Ov infection induces bile ducts and liver injury, leading to periductal fibrosis (PDF), which consequently develops CCA in animal models [2,3]. Similarly, Ov infection and PDF are recognized as the CCA risk factors in human subjects [4]. CCA is the leading cause of mortality in Thailand, Laos, Vietnam, and Cambodia. Therefore, it is crucial to promptly detect cases of Ov infection and PDF individuals in order to promote CCA control measures. Currently, PDF and CCA are screened using abdominal ultrasonography by the radiologists. Comprehensive analyses of tumor markers in the serum of PDF subjects were conducted using mass spectrometry-based techniques [5]. However, simple biochemical techniques remain required to identify and validate these biomarkers for the purpose of screening individuals who are at high risk of developing CCA.

Parasitological and molecular techniques have been developed to detect parasite larvae or eggs and DNA in fecal samples. One of the most reliable methods for diagnosing Ov infection is the egg detection by formalin ethyl-acetate concentration technique (FECT) [6]. This procedure accurately quantifies Ov eggs in feces and is considered the "gold standard" for diagnosis of Ov infection. The FECT approach revealed limited analytical sensitivity. Quantification of Ov eggs carries a technical error risk due to variations among microscopists. Ov eggs are similar in size to those of minute intestinal flukes (MIFs), requiring fresh feces samples for accurate results. As an alternative, PCR-based approaches were developed to detect parasite DNA in fecal samples using molecular techniques [7–9]. These methods exhibit high diagnostic sensitivity and accuracy when applied in an Ov-infected animal model. However, there are numerous components in human feces that can inhibit DNA extraction and PCR reactions; they may not be appropriate for all patients [10,11].

We have previously developed immunological methods using ELISA-based techniques to detect Ov antigens. The anti-Ov monoclonal antibody-producing hybridoma cell lines were induced and developed in-house by Worasith *et al.* [12]. The monoclonal antibody-based detection and quantitation of Ov antigens in urine samples showed much greater diagnostic sensitivity and specificity than the current gold standard method [12,13]. Therefore, this antigen detection method can serve as a powerful and efficient tool for the screening, management, and eradication of this tumorigenic parasite.

The in-house monoclonal antibody (Thai patent No. 2872) has been used for the detection and quantification of opisthorchiasis in animal and human studies [12,13]. The monoclonal antibody exhibited complete specificity for urine samples from

subjects infected with Ov (81%), with minimal cross-reactivity observed in urine samples from *S. stercoralis* infections (3.57%) and hookworm infections (10%) [12]. The antigen detection method was converted from ELISA to an immuno-chromatographic test, facilitating its use as a rapid diagnostic tool for opisthorchiasis in Thailand and other neighboring endemic countries [13].

In this study, the target protein from adult Ov that reacted with the monoclonal antibody was identified, and the recombinant protein was produced. The reactivity of the recombinant protein was evaluated and compared with the native Ov antigens by mAb-ELISA. Furthermore, the binding ability of target protein and mAb KKU505 was also performed using *in silico* analysis.

## Materials and methods

### Animal ethics

The protocols involving laboratory animals for the production of the adult Ov to prepare crude antigen were approved by the Animal Ethics Committee, Faculty of Medicine, Khon Kaen University (approval number AEKKU 98/2562). The protocols for monoclonal and polyclonal antibody productions were also approved by the Animal Ethics Committee, Faculty of Medicine, Khon Kaen University (approval number AEKKU 43/25620). Humane endpoints were implemented with daily health assessments of the animals to ensure their well-being. The procedure adhered strictly to the guidelines set forth by the National Research Council of Thailand for the Care and Use of Laboratory Animals.

### The crude Ov antigens preparation

Metacercariae of Ov were extracted from the naturally infected cyprinoid fish, which were obtained from an endemic area in northeastern Thailand. Fish were chopped into small pieces by an electrical blender, and the minced fish were enzymatically digested in a freshly prepared solution of 0.25% pepsin and incubated at 37°C for 1 h in a shaking water bath. After digestion, the digested material was filtered and washed several times. The Ov metacercaria were sorted from the final sediment and identified under a dissecting microscope. The metacercariae were pooled and administered orally via gastric intubation to five inbred male Syrian golden hamsters (*Mesocricetus auratus*) (50 cysts per animal), which yielded about 50% worm recovery according to the previous study [12]. The infected hamsters were housed and maintained at the animal center, Faculty of Medicine, Khon Kaen University. While the metacercariae dose was known to cause no effect on the well-being of the hamsters, the animals were monitored daily for any abnormal clinical, physiological, or behavioral signs, with criteria for humane endpoint intervention in place. No significant morbidity was observed over the 12-week infection period. The animals were then euthanized with isoflurane prior to removal of the liver to recover adult worms. The intact worms were cleaned and homogenized on ice and centrifuged at 14,000 rpm at 4°C for 30 min, and the supernatant obtained was stored at -20°C until required. The protein concentration of the crude somatic Ov antigen was determined using the BCA protein assay kit (Thermo Fisher Scientific, MA, USA).

### Monoclonal and polyclonal antibody production

The monoclonal antibody (mAb) clone KKU505, previously developed and validated by Worasith et al. [12] for mAb-ELISA detection of Ov antigen in urine, was used in this study. For this study, the mAb was produced in laboratory animals using two inbred male BALB/c mice, following standardized procedures. The mice were purchased from Nomura Siam International Company Limited and housed at the Northeast Laboratory Animal Center, Khon Kaen University. After a 7-day acclimatization period, they received an intraperitoneal injection of 0.78 g/mL pristane solution (SIGMA, St. Louis, USA) for 0.25 mL per animal. Seven days later, hybridoma cells ($1–3 \times 10^6$ cells/mL PBS per animal) were prepared and injected intraperitoneally. Ascitic fluid containing the mAb was collected 2–4 weeks post-injection by experienced animal technicians. Humane endpoint criteria were strictly observed, with daily monitoring for clinical, physiological, and behavioral

abnormalities. Once the endpoint criteria were met, euthanasia was performed within three hours. No animals died before reaching the predefined endpoint. The collected ascitic fluid was centrifuged at 2,000 rpm for 10 minutes, and the supernatant was aliquoted and stored at -80°C until use.

The rabbit IgG polyclonal antibody against Ov was generated by immunizing two New Zealand white rabbits (*Oryctolagus Cuniculus*) with crude Ov antigen. After a week of acclimatization, the animals were first immunized with a subcutaneous injection of 100 µg of Ov antigen suspended in complete Freund's adjuvant (CFA, SIGMA® (St. Louis, USA). Four weeks later, the second immunization was performed using 100 µg of Ov antigen mixed with incomplete Freund's adjuvant (IFA, SIGMA®, St. Louis, USA), followed by a booster injection of the same antigen formulation two weeks after that. At week 10 after the initial immunization, the rabbits were euthanized using isoflurane, and blood was collected via cardiac puncture by the experienced animal technician. No animals had abnormal physiological signs before the schedule for euthanasia. The serum was separated and stored at -20°C for future use.

### Sodium dodecyl sulfate polyacrylamide gel electrophoresis (SDS-PAGE)

The protein sample was mixed with SDS loading sample buffer, which had a final concentration of 62.5 mM Tris-HCl pH 6.8, 20% glycerol, 2% SDS, and 5% β-mercaptoethanol. The mixture was then boiled to 100°C for 10 min and subjected to electrophoresis on either an 10% or 8% or 6% SDS-PAGE gel. For the cut-off target size of protein bands, electrophoresis was performed at 40 mA (per gel) for 40 min, or until the bromophenol blue dye reached the bottom of the gel, or by following a pre-stained protein ladder. SDS-PAGE gel was stained with Coomassie Brilliant Blue (CBB) R-250 solution containing 10% acetic acid with 25% methanol for 1 h. Gels were de-stained in 8% acetic acid for 1 h or until the background was cleared.

### Western blot analysis

After electrophoresis, proteins were transferred from the SDS-PAGE gel to the PVDF membrane (Immobilon®, Millipore, MA, USA). The electroblotting was carried out at 30 V, 0.1 A, overnight at room temperature with a Bio-Rad Transblot apparatus and transfer buffer (25 mM Tris, 192 mM Glycine, 20% v/v methanol, pH 8.3). After transfer, the PVDF membrane was incubated in 5% skim milk with TBS buffer for 1 h to block non-specific protein binding. Following a blocking step, the membrane was probed with a mAb that was obtained from ascitic fluid (1:1,000 in 1% skim milk) or culture media (1:100 in 1% skim milk) for 1 h on the shaker. Then, the membrane was washed twice with 0.05% Tween-20 in TBS buffer (TTBS) and washed with TBS buffer for 5 min on the shaker. After that, a secondary antibody, 1:1,000 in 1% skim milk of goat anti-mouse IgG/IgM conjugated-HRP (Sigma-Aldrich, USA), was added and incubated for 1 h on the shaker. Following a further washing process, a chemiluminescence substrate specific to the HRP enzyme was applied onto the membrane. The HRP enzyme then modifies the substrate, resulting in the emission of light as detected by the Chemiluminescence Imaging Systems (Amersham™ Imager 600).

### Digestions of O-glycans and N-glycans

Digestions of O-glycans and N-glycans were performed using O-glycosidase and PNGase F protocols from the New England BioLabs® Inc, MA, USA. To denature glycoproteins, the reaction mixture containing 10 µg of glycoprotein sample, 1 µL of glycoprotein denaturing buffer (10×), and deionized water was mixed in a total reaction volume of 10 µL in a microtube. The reaction mixture was heated at 100°C for 10 min. After that, 2 µL of GlycoBuffer-2 (10×), 2 µl of NP-40, and glycosidase enzyme (1 µL of O-glycosidase and 2 µL of neuraminidase or 1 µL of PNGase F from New England BioLabs® Inc, MA, USA) were added, and deionized water was added for up to 20 µL of total volume. The reaction mixture was incubated at 37°C for 1 h. The results of digestion were checked using SDS-PAGE and western blot analysis.

## Liquid chromatography coupled with tandem mass spectrometry (LC-MS/MS)

Expected protein targets of the mAb were identified using liquid chromatography tandem mass spectrometry (LC-MS/MS). Protein bands of interest were cut from the SDS-PAGE gel (CBB-stained gel) and placed into a 1.5 mL microcentrifuge tube. The sample in gel was digested with trypsin. After digestion, the samples were cleaned up using C18 Zip Tips™ (Millipore) and analyzed with a nano-liquid chromatography system (EASY-nLC II, Bruker, MA, USA) coupled to an ion trap mass spectrometer (Amazon Speed ETD, Bruker) combined with an ESI nanosprayer. The ESI-TRAP instrument was calibrated at the m/z range 50−3000 using an internal calibration standard. The autosampler loaded a 3 µL sample volume at a flow rate of 500 nL/min onto an EASY-Column, 10 cm, ID 75 µm, 3 µm, C18-A2 (Thermo Scientific, MA, USA). DI water was present in mobile phase A, while acetonitrile and 0.1% formic acid were present in mobile phase B. The gradient was 5–35% B for 50 min, and then 80% B for 10 min. The experiment was conducted using tryptic digest (50 fmol) of bovine serum albumin (BSA) as the standard control. The ion trap device was operated by the HyStar v.3.2 software package from Bruker Daltonics. Compass Data Analysis v.4.0 was used to examine LC-MS/MS spectra. For additional searching in the MASCOT application, compound lists were exported as Mascot generic files (mgf).

## Bioinformatics and sequence analysis

Amino acid sequences from LC-MS/MS analysis were searched for homology versus gene and protein in public databases using a Blast search at https://blast.ncbi.nlm.nih.gov/Blast.cgi [14]. Sequences were edited and analyzed for coding regions and restriction maps using BioEdit.

## Recombinant protein production

The target clones, full-length Ov myosin heavy chain, were annotated from the genome sequence of Ov (GenBank acc. XM_009176183) (full-length Ov myosin heavy chain, spanning nt. 120–5,918 encoding 1,932 aa and partial Ov myosin head, spanning nt. 120–2,256 encoding 712 aa). The targeted sequences of the myosin heavy chain were synthesized and cloned into the pET-15b vector using the Nde I site (CATATG) (GeneScript Centennial Ave., Piscataway, NJ, USA). The *Escherichia coli* strain TOP10 was used to replicate the recombinant sequences. The target clones were verified by PCR with T7 promoter primers. The recombinant plasmids were transformed into *E. coli* strain BL21 (DE3) (Novagen Madison, WI, USA) for target protein expressions. The foreign plasmids were transfected through the cell wall by the heat shock method at 42°C and the transformed cells were spread on LB agar supplemented with ampicillin (1 µg/mL) and incubated at 37°C with horizontal shaking at 200 rpm overnight. Single colonies were picked and grew in 200 mL LB broth with ampicillin (1 µg/mL) at 37°C until the OD600 was approximate 0.6. After that, the recombinant protein expressions were induced by adding 0.1 mM of isopropyl β-D-1-thiogalactopyranoside (IPTG) and incubated at 37°C overnight with shaking. After cell growth, the cells were aliquoted and collected by centrifugation at 5,000 *g* for 15 min at 4°C. The supernatant was removed, and cell pellets were kept at -20 °C until used.

 **Total protein extraction from *E. coli* strain BL21 for SDS-PAGE.** The bacteria cell pellets were mixed with 500 µL of SDS loading sample buffer. The mixtures were then boiled to 100°C for 10 min. Then, the tubes were centrifuged at 12,000 rpm for 5 min. The supernatants were collected and subjected to SDS-PAGE using 8% SDS-PAGE gel as above mentioned.

 **Total protein extraction from *E. coli* strain BL21 for enzyme-linked immunosorbent assay.** The bacteria cell pellets were mixed with 5 mL of PBS and sonicated for 5 min on ice. Then, the tubes were centrifuged at 12,000 rpm for 5 min. The supernatants were collected and kept at -20°C until used.

## Isolation of the full-length myosin heavy chain recombinant protein by serial extraction

The bacteria cell pellet (the full-length myosin heavy chain expression) was added by 2 mL of the extraction buffer 1 (50 mM NaH$_4$PO$_4$, 75 mM NaCl, pH 8.2) and sonicated for 5 min on ice. Then, the tube was centrifuged and the

supernatant was collected as supernatant 1 (S1, Fig 3A). The pellet was further washed again with 2 mL of extraction buffer 1, sonicated, centrifuged, and the supernatant was removed. After washing, the pellet was re-suspended in the extraction buffer 2 (50 mM $NaH_4PO_4$, 150 mM NaCl, pH 8.2) and sonicated for 5 min on ice. Then, the tube was centrifuged, and the supernatant was collected as supernatant 2 (S2, Fig 3A). After that, the pellet was re-suspended in the extraction buffer 3 (20 mM $NaH_4PO_4$, 500 mM NaCl, 2M thiourea, 7M urea, pH 8.0) and mixed by a vortex mixer. The mixture was labeled as supernatant 3 (S3, Fig 3A). Then, the supernatant 3 was mixed with 100% ammonium sulfate, to make up a final concentration of 30% ammonium sulfate for salting out the recombinant protein. The mixture was centrifuged at 10,000 rpm at 4°C for 10 min. The supernatant was removed, and the pellet was washed 3 times with acetone. Subsequently, the pellet was re-dissolved at 4°C in PBS overnight, and the mixture was labeled as supernatant 4 (S4). After that, SDS-PAGE was used to separate supernatant 1–3, and S4 or purified recombinant full-length Ov myosin heavy chain (rMHC) was used for the enzyme-linked immunosorbent assay. The leftover samples were kept at -20°C.

## Capture enzyme-linked immunosorbent assay (Capture-ELISA)

The protocol for capture-ELISA was similar to those described previously using the same mAb clone (KKU505) and rabbit anti-Ov IgG [12,13,15]. Polystyrene microtiter plates (NUNC, Roskilde, Denmark) were sensitized overnight at 4°C with 5 µg/mL of mAb of a clone KKU505. Plates were washed three times with 0.05% Tween 20 in PBS (pH = 7.4) (PBST), and uncoated sites were blocked with 5% skim milk in carbonate buffer, pH 9.6. After incubation for 1 h at 37°C, spiked serially diluted purified recombinant myosin heavy chain (rMHC) in PBS was added, and the plates were incubated at 37°C for 2 h. After five washes with PBST, 10 µg/mL of rabbit anti-Ov IgG was added to each plate and then incubated at 37°C for 1 h. Following the addition of biotin goat anti-rabbit IgG (dilution 1:4,000) (Invitrogen, CA, USA), the plates were incubated at 37°C for 1 h. After washing three times, streptavidin-horseradish peroxidase conjugate (dilution 1:5,000) (GE Healthcare, Buckinghamshire, UK) was added, and the plates were incubated at 37°C for 1 h. The plates were washed, and the substrate solution (o-phenylenediamine hydrochloride) (Sigma, St. Louis, MO, USA) was added and incubated for 20 min in the dark at room temperature. The enzyme reaction was stopped with 2 M sulfuric acid, and the optical densities (OD) were read spectrophotometrically at 492 nm with an ELISA reader (Tecan Sunrise Absorbance Reader, Austria). The equilibrium dissociation constant ($K_D$) is used to represent the binding affinity of molecules. $K_D$ between monoclonal antibody (mAb KKU505) and the targeted proteins (crude Ov antigens and rMHC) were analyzed by GraphPad Prism 9 software from OD492 values.

## Structural modeling and molecular docking

The amino acid sequences of mAb KKU505 and the Ov myosin head domain (residues 1–770) of the myosin heavy chain were obtained from Siripanthong et al. [16]. The 3D structures of both proteins were predicted using ChimeraX AlphaFold [17]. Based on the local distance difference test (IDDT) score, the best model of each protein was selected for molecular docking using the ZDOCK web server (https://zdock.umassmed.edu/) [18,19]. The docked structure with the highest docking score was selected for further study. Note that the tail of the myosin heavy chain (residue771–800) was removed from the docked complex to reduce computational time. The protonation state of the protein-protein complex was predicted at pH 7.0 using the PROPKA web server [20]. The ff14SB force field parameter [21] was applied for the protein-protein complex. The $Na^+$ ions were added to neutralize the system. Then, the complex was solvated using the TIP3P water model in an octahedral box [22]. After that, the system was minimized using the steepest descent and the conjugated gradient methods.

## Molecular dyvnamics (MD) simulation

MD simulation of the complex of mAb KKU505 and the Ov myosin head domain was performed using the AMBER22 software package [23]. Firstly, the system was heated up from 100 to 310 K for 100 ps with a canonical ensemble (NVT) with a time step of 2 fs. Secondly, the water molecules and the complex were equilibrated for 100 and 500 ps, respectively. Finally,

the system was simulated for 100 ns under the isothermal-isobaric ensemble (*NPT*) at 1 atm and 310 K with a time step of 2 fs. The pressure and the temperature of the system were maintained using the Berendsen barostat [24] (pressure-relaxation time = 1 ps) and the Langevin thermostat [25] (collision frequency = 2 ps$^{-1}$), respectively. The SHAKE method [26] was applied to constrain all bonds involving hydrogens. The particle-mesh Ewald method [27] was used to manage the long-range electrostatic interactions. After completing the MD simulation, the MD trajectories were gathered for analysis of the root-mean-square displacement (RMSD) using the CPPTRAJ module in the AMBER22 program [23]. The intermolecular interactions between KKU505 Fab and the myosin head were evaluated using the PDBsum web server.

### B cell epitope prediction

The full-length Ov myosin heavy chain protein was annotated from the genome sequence of Ov (GenBank acc. XM_009176183) and was used to predict B cell epitopes using EpiGraph (http://epigraph.kaist.ac.kr/) developed by Choi and Kim. B cell epitope prediction was performed by capturing the spatial clustering property of the epitopes using a graph attention network [28].

## Results

### Detection and identification of the targeted proteins recognized by monoclonal antibody

Using SDS-PAGE, the crude Ov antigens of adult worms were separated. The CBB-stained 6% gel and the western blot result were shown in Fig 1A-B. The primary antibody applied in these assays was a mAb derived from KKU505 cells (mAb KKU505). As illustrated in Fig 1A-B, the candidate protein band was detected at approximately over 180 KDa on 6% acrylamide gel. The gel stained with CBB for the Ov crude antigens (Fig 1A) revealed two protein bands that were clearly stained. The arrow indicated the selected protein band, which was analyzed further using LC-MS/MS. Ov-crude antigens treated with N-glycosidase, O-glycosidase, and N- and O-glycosidases did not affect the immunoreactivity patterns (Fig S1), indicating that the targeted epitopes are not associated with glycan parts.

The results of the LC-MS/MS analysis for the selected protein band are presented in S1 File. The protein was identified as the myosin head of Ov with protein sequence coverage of 21% of the OON15278 sequence. The myosin head contains a monoisotopic mass (Mr) of 222.414 KDa. Myosin head domain is a part of myosin heavy chain. Thus, the myosin heavy chain of the parasite was selected for further characterization.

### Recombinant Ov myosin protein production and characterization

The full mRNA sequence of Ov myosin heavy chain was obtained from the shotgun genome sequence of the Ov genome project submitted to GenBank (GenBank acc. XM_009176183) [29].

The target clones were verified by PCR with T7 promoter primers (Fig S2). Around the 75 KDa band of the partial Ov myosin head and around the 200 KDa protein band of the full-length Ov myosin heavy chain were detected in *E. coli* BL21 as shown in Fig 2A. The LC-MS/MS results of the ~200 KD (S2 File) and ~75 KD (S3 File) protein bands showed that they were indeed the myosin head of Ov. In addition, as shown in Fig 2B, the recombinant protein of the full-length Ov myosin heavy chain reacted with the monoclonal antibody as determined by capture-ELISA, whereas the partial Ov myosin head did not react with it. Since the crude protein isolated from *E. coli* BL21 transfected with the full-length Ov myosin heavy chain exhibited a lower intensity (OD492) in the capture-ELISA in comparison to the crude Ov antigens, full-length Ov myosin heavy chain protein purification is necessary to increase immunoreactivity.

### Isolation, purification, and immunoreactivity of recombinant full-length Ov myosin heavy chain protein

The recombinant full-length Ov myosin heavy chain protein (rMHC) was purified by the serial extraction as described in the Materials and Methods section. The single band of recombinant full-length Ov myosin heavy chain protein in PBS was

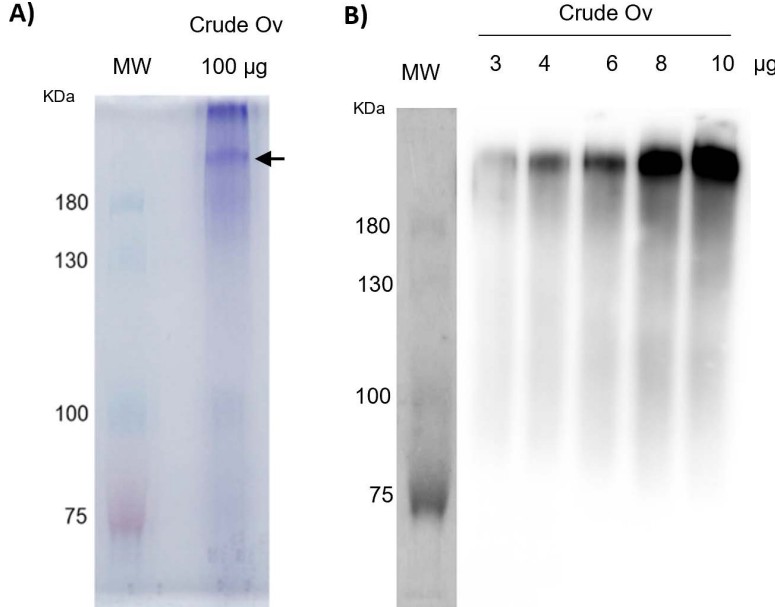

**Fig 1. Separation of Ov crude antigens using SDS-PAGE and western blot analysis. (A)** CBB staining of 100 µg of Ov crude antigens separated using 6% acrylamide gels. **(B)** Western blot analysis of 3-10 µg of Ov crude antigens using mAb KKU505 clone as the primary antibody. Arrow indicates the selected protein for LC-MS/MS analysis. MW = standard protein molecular weights, KDa = kilodalton, and Crude Ov = Ov crude extract.

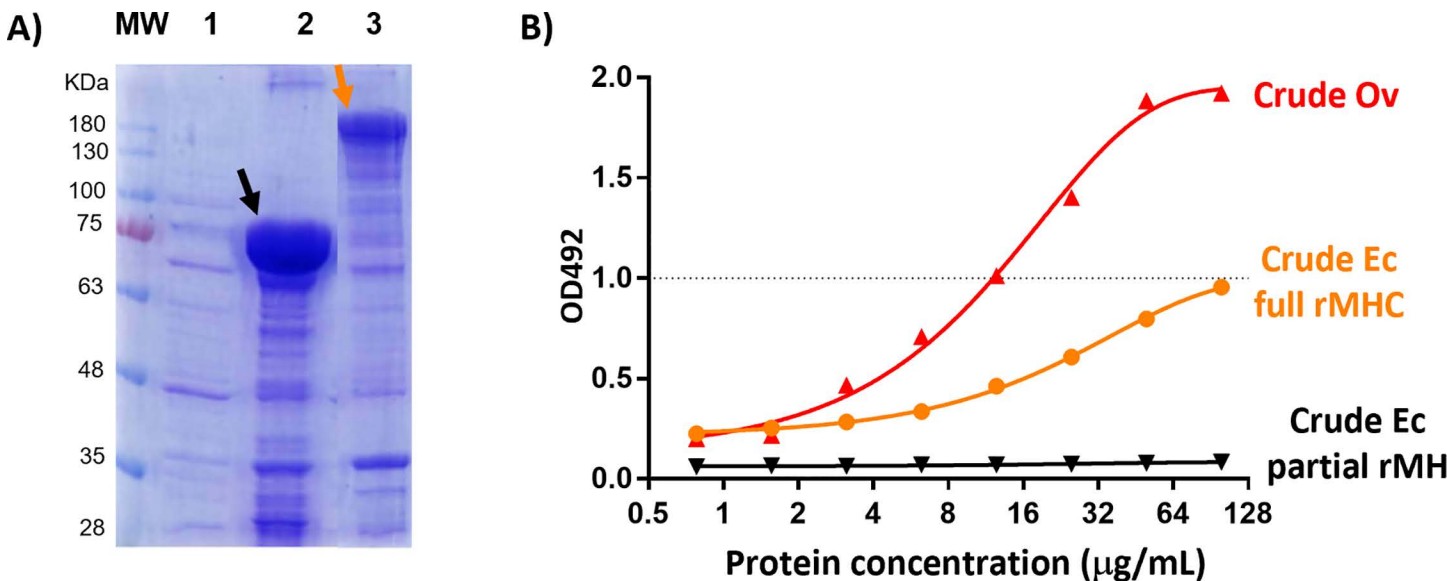

**Fig 2. Recombinant Ov-myosin protein production in *E. coli* BL21. (A)** CBB staining of 8% acrylamide gel of crude extracted from *E. coli* BL21 (1), *E. coli* BL21 with partial Ov-myosin head (2), and *E. coli* BL21 with full Ov-myosin heavy chain transfection (3). The black arrow indicates the partial Ov-myosin head protein, and the orange arrow indicates the full Ov-myosin heavy chain protein. **(B)** Capture-ELISA results of proteins crude extracts from *O. viverrini* (Crude Ov), *E. coli* BL21 with partial Ov-myosin head expression (Crude Ec partial rMH), and *E. coli* BL21 with full Ov-myosin heavy chain expression (Crude Ec full rMHC).

obtained after the final step of purification (Fig 3A). Immunoreactivity of rMHC was compared to that of crude Ov antigens using the capture-ELISA (Fig 3B). The immunoreactivity of rMHC was higher than that of crude Ov antigens at the concentration range from 0.006 to 12.5 µg/mL. At protein concentrations exceeding 25 µg/mL, the reactivities of rMHC and crude Ov antigens were comparable (Fig 3B). The raw data (OD492 values) from Fig 3B were utilized to ascertain the $K_D$ values between the mAb and crude Ov (5.298 µg/mL), as well as between the mAb and rMHC (1.329 µg/mL), using GraphPad Prism 9 software. The results indicated that rMHC exhibited a higher affinity for the mAb compared to crude Ov antigens.

## Molecular modeling of KKU505 Fab in complex with Ov myosin head

To elucidate the binding mechanism of KKU505 Fab against the Ov myosin head domain, molecular docking was first conducted using the ZDOCK Server. As shown in Fig 4A, both chains A and B of KKU505 Fab could bind to the myosin head domain with the ZDOCK score of 942.182. To evaluate further the structural stability of the KKU505 Fab/myosin head domain complex in an aqueous environment, MD simulation was performed. As shown in Fig 4B, the protein-protein complex reached the equilibrium state after 20 ns with the RMSD fluctuation of 3.5 to 5.0 Å, suggesting the high stability of the complex in water solution. The protein-protein interactions between KKU505 Fab and the Ov myosin head domain were predicted using the PDBsum. The results (Fig 4C) showed that chain A of KKU505 Fab can form 33 non-bonded contacts, whereas chain B of KKU505 Fab can form 58 non-bonded contacts with the Ov myosin head domain. In addition, seven hydrogen bonds were detected between chain B of KKU505 Fab and the Ov myosin head domain: (i) Lys275 and Asn601 (two bonds), (ii) Pro274 and Asn601, (iii) Tyr316 and Glu621, (iv) Tyr319 and Glu621, (v) Gln270 and Lys398, and (vi) Asn268 and Lys627. Taken together, these data indicate that KKU505 Fab could interact with the Ov myosin head domain with high structural stability. Additionally, B cell epitope prediction confirmed that B cells might recognize the amino acid residues of the Ov myosin head domain from 461 to 628 (S4 File).

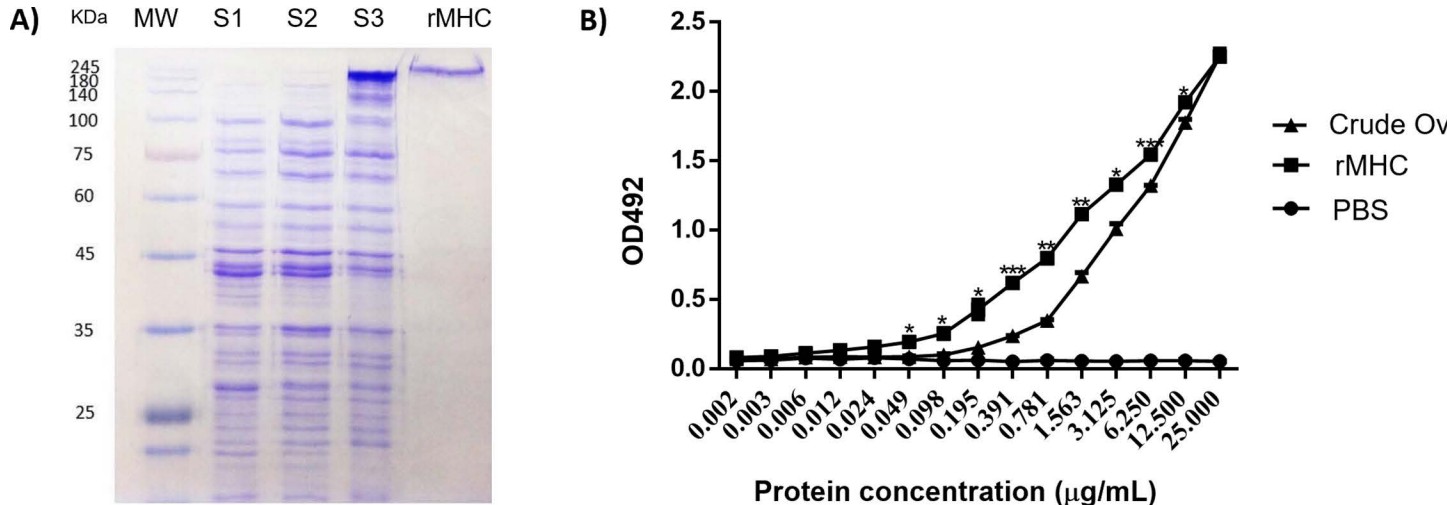

**Fig 3. Isolation, purification, and immunoreactivity of recombinant full-length Ov myosin heavy ch ain. (A)** CBB staining of 10% acrylamide gel of proteins from supernatant 1 (S1), supernatant 2 (S2), supernatant 3 (S3), and recombinant Ov myosin heavy chain (rMHC) or supernatant 4 obtained from the serial extraction of the full-length Ov myosin heavy chain expression in *E. coli* BL21. **(B)** Capture-ELISA of crude Ov and rMHC curves. MW = standard protein molecular weights, KDa = kilodalton, Crude Ov = Ov crude extract, rMHC = recombinant full-length Ov myosin heavy chain. *$P$ < 0.05, **$P$ < 0.01, and ***$P$ < 0.001 compare to crude Ov.

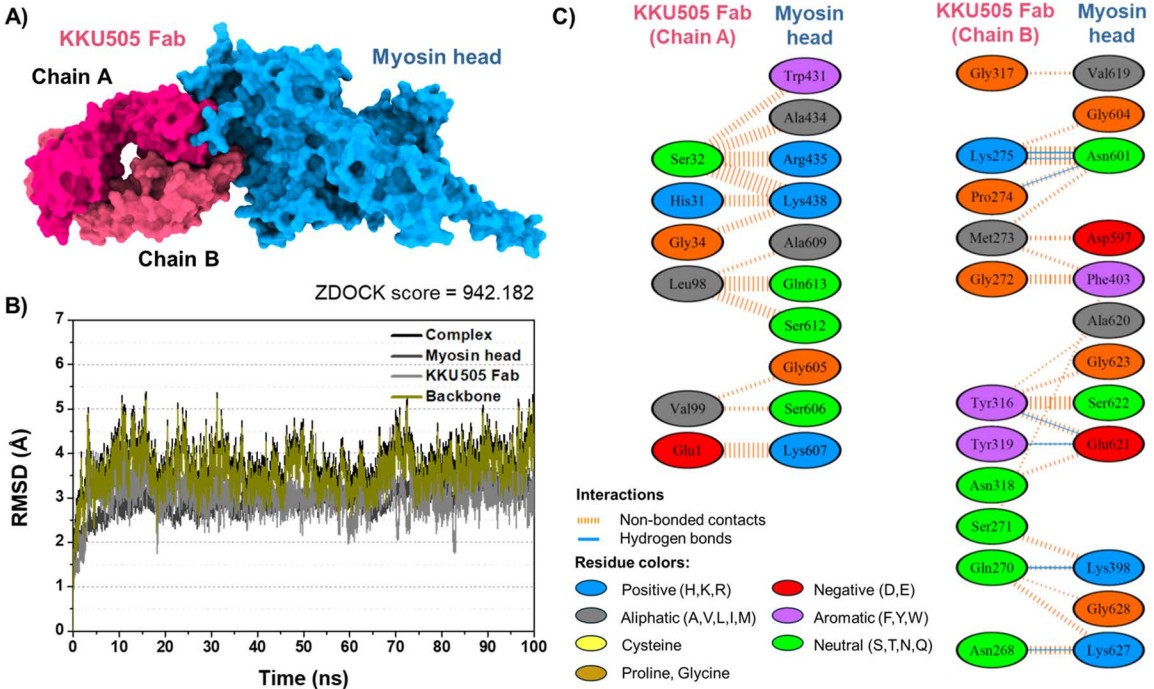

**Fig 4. Molecular modeling of KKU505 Fab in complex with Ov myosin head domain. (A)** Docked structure of KKU505 Fab in complex with myosin head. **(B)** Time evolution of RMSD. **(C)** Protein-protein interaction profile between KKU505 Fab and myosin head domain.

## Discussion

Several muscle-associated proteins, including myosin, actin, paramyosin, and tropomyosin, have been identified in trematodes. Myosin has been detected in the somatic muscle tissues, as well as in the muscles of the oral sucker and uterus of *Fasciola hepatica* [30], and on the tegument of *Schistosoma mansoni* [31]. Moreover, myosin is a key target for vaccine candidates and praziquantel-based chemotherapy in opisthorchiasis and schistosomiasis [32–34].

In this study, we successfully identified and characterized a novel 222 KDa Ov protein, myosin heavy chain (MHC), which is a target molecule of our in-house mAb against Ov. The crude Ov antigens were extracted from the adult worms, and the specificity of the protein pattern was assessed using western blot with an in-house mAb derived from KKU505 cell culture media and ascitic fluid . Since immunoreactivity patterns were unchanged after glycosidase treatments of Ov-crude antigens, the antigenic epitope(s) of the target protein do not belong to the carbohydrate part. The LC-MS/MS analysis of the selected protein band was identified as the myosin head domain of Ov myosin heavy chain.

Myosin head is the head domain of myosin heavy chain that facilitates the contraction of the parasite muscles, thus contributing to the diverse muscular activities exhibited by flukes. Moreover, *in silico* analysis confirmed the binding capacity between the monoclonal antibody and the Ov myosin head domain. The molecular docking results demonstrated that the Ov myosin head domain employs amino acid residues from Asn268 to Gly628 for interaction with the mAb KKU505. The amino acid residues from 461 to 628 were also predicted to have an antigenicity to B cells. The recombinant full-length Ov myosin heavy chain and partial Ov myosin head were produced. The partial Ov myosin head was constructed exclusively in the N-terminal region (712 amino acids) to enhance the protein production and preparation efficiency. Nevertheless, it demonstrated negative reactivity with the monoclonal antibody. These results suggest that the partial myosin head protein expression may lack proper conformation, leading to diminished immunoreactivity. By contrast, the binding efficacy of full-length Ov myosin heavy chain (rMHC) and Ov crude antigens were compared in the capture-ELISA

at the same protein concentration. The results indicated that the immunoreactivities of rMHC against the in-house mAb was comparable to that of the crude Ov antigens. Therefore, the rMHC protein, which is reactive to the mAb KKU505, can serve as a novel positive control, replacing the Ov crude antigen in mAb-ELISA-based as well as immunochromatographic antigen detection. Using rMHC reduces the need for laboratory animals to produce Ov worms, aligning with the 3R principles of animal ethics—Replacement, Reduction, and Refinement.

This study identifies several key areas for further investigation. First, a confirmatory study is needed to determine whether the antigen detected in the urine or feces of Ov-infected individuals includes the Ov myosin head. Second, further assessment is required to determine whether the rMHC described in this study shares biochemical properties with the native myosin heavy chain molecule, as multiple isoforms may arise due to post-translational modifications and protein preparation. Lastly, the isolation and characterization of target molecules in the urine of Ov-infected individuals, as recognized by the current mAb, warrant further study. Clarifying the nature and composition of these antigens is crucial for accurately interpreting Ov antigen profiles in curative treatment. Moreover, understanding how Ov-specific antigens originate from worms in the biliary system is essential for improving Ov diagnostics and associated pathology.

## Conclusion

Our study focused on identifying and characterizing the myosin heavy chain of Ov adult worm which reacted to a monoclonal antibody (mAb KKU505) against *Opisthorchis viverrini*. The availability of the recombinant Ov myosin heavy chain (rMHC) protein as a substitute for crude Ov antigen reduces costs and minimizes the need for laboratory animals for adult worm production. Additionally, mapping the myosin epitopes recognized by mAb lays the groundwork for further research in identifying urinary antigens for opisthorchiasis diagnosis.

## Supporting information

**S1 Fig. Western blot analysis of Ov crude antigens after treatments with glycosidases.** The proteins were separated using 6% acrylamide gels of SDS-PAGE follow by western blot analysis with the mAb KKU505 clone as a primary antibody. MW = standard protein molecular weights and KDa = kilodalton.
(TIF)

**S2 Fig. The PCR products of Ov myosin overexpression in *E. coli* strain BL21 systems were detected using agarose gel electrophoresis using T7 promoter primers.** The 2,560 bp PCR products represents partial Ov myosin head inserted sequence vector and 5,853 bp PCR products represent full-length myosin heavy chain inserted sequence vector. NTC = Negative control.
(TIF)

**S1 File. LC-MS/MS analysis data sheet of Ov myosin protein from Ov crude antigens.**
(PDF)

**S2 File. LC-MS/MS analysis data sheet of the recombinant full-length Ov myosin heavy chain protein from *E. coli* strain BL21 crude extract.**
(PDF)

**S3 File. LC-MS/MS analysis data sheet of the recombinant partial Ov myosin head domain from *E. coli* strain BL21 crude extract.**
(PDF)

**S4 File. B cell epitope prediction of Ov myosin head and tail domains.**
(PDF)

## Acknowledgments

We would like to acknowledge Prof. Yukifumi Nawa, for editing the MS via Publication Clinic KKU, Thailand.

## Author contributions

**Conceptualization:** Anchalee Techasen, Thewarach Laha, Paiboon Sithithaworn, Raynoo Thanan.

**Data curation:** Chanika Worasith, Duangkamon Muengsaen, Jiraprapa Ponglong.

**Formal analysis:** Anchalee Techasen, Chanika Worasith, Duangkamon Muengsaen, Jiraprapa Ponglong, Panupong Mahalapbutr, Napat Kongtaworn, Thanyada Rungrotmongkol, Sujittra Chaiyadet, Chadamas Sakonsinsiri, Raynoo Thanan.

**Funding acquisition:** Anchalee Techasen, Paiboon Sithithaworn, Raynoo Thanan.

**Investigation:** Anchalee Techasen, Chanika Worasith, Duangkamon Muengsaen, Jiraprapa Ponglong, Sujittra Chaiyadet.

**Methodology:** Anchalee Techasen, Thewarach Laha, Paiboon Sithithaworn, Raynoo Thanan.

**Project administration:** Paiboon Sithithaworn, Raynoo Thanan.

**Resources:** Kanoknan Khongsukwiwat, Phattharaphon Wongphutorn, Chompunoot Wangboon, Chutima Homwonk, Sujittra Chaiyadet, Sutas Suttiprapa, Paiboon Sithithaworn, Raynoo Thanan.

**Software:** Panupong Mahalapbutr, Napat Kongtaworn, Thanyada Rungrotmongkol.

**Supervision:** Anchalee Techasen, Thewarach Laha, Paiboon Sithithaworn, Raynoo Thanan.

**Validation:** Anchalee Techasen, Chanika Worasith, Duangkamon Muengsaen, Jiraprapa Ponglong, Napat Kongtaworn, Thanyada Rungrotmongkol, Sujittra Chaiyadet, Thewarach Laha, Sutas Suttiprapa, Chadamas Sakonsinsiri, Paiboon Sithithaworn, Raynoo Thanan.

**Visualization:** Anchalee Techasen, Chanika Worasith, Duangkamon Muengsaen, Jiraprapa Ponglong, Panupong Mahalapbutr, Napat Kongtaworn, Thanyada Rungrotmongkol, Raynoo Thanan.

**Writing – original draft:** Anchalee Techasen.

**Writing – review & editing:** Chanika Worasith, Duangkamon Muengsaen, Jiraprapa Ponglong, Panupong Mahalapbutr, Napat Kongtaworn, Thanyada Rungrotmongkol, Kanoknan Khongsukwiwat, Phattharaphon Wongphutorn, Chompunoot Wangboon, Chutima Homwonk, Sujittra Chaiyadet, Thewarach Laha, Sutas Suttiprapa, Chadamas Sakonsinsiri, Paiboon Sithithaworn, Raynoo Thanan.

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
