## [Decision Letter · Decision Letter 0]

26 Dec 2024

PONE-D-24-36944Identification and characterization of a target antigen recognized by the monoclonal antibody against Opisthorchis viverriniPLOS ONE

Dear Dr. Thanan,

Thank you for submitting your manuscript to PLOS ONE. After careful consideration, we feel that it has merit but does not fully meet PLOS ONE’s publication criteria as it currently stands. Therefore, we invite you to submit a revised version of the manuscript that addresses the points raised during the review process.

**Double check the MS for English usage, and typos.****Demonstrate the antigen recognition pattern, and the usefulness in Ov and non-Ov samples.****Clear the points raised by the Reviewers, especially the additional experiments required. **

 Please submit your revised manuscript by Feb 09 2025 11:59PM. If you will need more time than this to complete your revisions, please reply to this message or contact the journal office at plosone@plos.org . Please include the following items when submitting your revised manuscript:

We look forward to receiving your revised manuscript.

Kind regards,

Marcello Otake Sato, Ph.D., D.V.M.

Academic Editor

PLOS ONE

**Journal Requirements:**

2. To comply with PLOS ONE submissions requirements, in your Methods section, please provide additional information regarding the experiments involving animals and ensure you have included details on (a) methods of sacrifice, (b) methods of anesthesia and/or analgesia, and (c) efforts to alleviate suffering."

This research was supported by the Fundamental Fund of Khon Kaen University from the National Science, Research and Innovation Fund (NSRF) and National Research Council of Thailand (NRCT) via Cholangiocarcinoma Research Institute, Khon Kaen University.

This research was supported by the National Research Council of Thailand (NRCT) via Cholangiocarcinoma Research Institute, Khon Kaen University and the Fundamental Fund of Khon Kaen University from the National Science, Research and Innovation Fund (NSRF). We would like to acknowledge Prof. Yukifumi Nawa, for editing the MS via Publication Clinic KKU, Thailand. 

This research was supported by the Fundamental Fund of Khon Kaen University from the National Science, Research and Innovation Fund (NSRF) and National Research Council of Thailand (NRCT) via Cholangiocarcinoma Research Institute, Khon Kaen University.

5. In the online submission form, you indicated that The data that support the findings of this study are available from the corresponding author upon reasonable request.

6. PLOS requires an ORCID iD for the corresponding author in Editorial Manager on papers submitted after December 6th, 2016. Please ensure that you have an ORCID iD and that it is validated in Editorial Manager. To do this, go to ‘Update my Information’ (in the upper left-hand corner of the main menu), and click on the Fetch/Validate link next to the ORCID field. This will take you to the ORCID site and allow you to create a new iD or authenticate a pre-existing iD in Editorial Manager.

7. We note that you have included the phrase “data not shown” in your manuscript. Unfortunately, this does not meet our data sharing requirements. PLOS does not permit references to inaccessible data. We require that authors provide all relevant data within the paper, Supporting Information files, or in an acceptable, public repository. Please add a citation to support this phrase or upload the data that corresponds with these findings to a stable repository (such as Figshare or Dryad) and provide and URLs, DOIs, or accession numbers that may be used to access these data. Or, if the data are not a core part of the research being presented in your study, we ask that you remove the phrase that refers to these data.

8. PLOS ONE now requires that authors provide the original uncropped and unadjusted images underlying all blot or gel results reported in a submission’s figures or Supporting Information files. This policy and the journal’s other requirements for blot/gel reporting and figure preparation are described in detail at https://journals.plos.org/plosone/s/figures#loc-blot-and-gel-reporting-requirements and https://journals.plos.org/plosone/s/figures#loc-preparing-figures-from-image-files. When you submit your revised manuscript, please ensure that your figures adhere fully to these guidelines and provide the original underlying images for all blot or gel data reported in your submission. See the following link for instructions on providing the original image data: https://journals.plos.org/plosone/s/figures#loc-original-images-for-blots-and-gels.   

**Additional Editor Comments:**

The study by Thanan and cols. aimed to support an important area in opisthorchiasis, improving the diagnosis of the parasite in terms of sensitivity and specificity. Despite the merit of the study, there are several points to be addressed in order to determine the reliability of the findings, such as monoclonal KKU505 antigen recognition pattern, and the usefulness in Ov and non-Ov samples.

Reviewers' comments:

Reviewer's Responses to Questions

**Comments to the Author**

1. Is the manuscript technically sound, and do the data support the conclusions?

Reviewer #1: Partly

Reviewer #2: Partly

2. Has the statistical analysis been performed appropriately and rigorously? 

Reviewer #1: I Don't Know

Reviewer #2: N/A

3. Have the authors made all data underlying the findings in their manuscript fully available?

Reviewer #1: No

Reviewer #2: Yes

4. Is the manuscript presented in an intelligible fashion and written in standard English?

Reviewer #1: No

Reviewer #2: Yes

5. Review Comments to the Author

**Reviewer #1:**  The manuscript entitled "Identification and characterization of a target antigen recognized by the monoclonal antibody against Opisthorchis viverrini" by Techasen and coworkers had been reviewed. Generally, the study in the manuscript dealt with an interesting topic. The study format was well designed and the data are valuable for publication. I do feel that it has merit but does not fully meet the publication standard as it currently stands. The authors can submit an improved version of the manuscript that addresses the points raised as follows as well as in the attached review report.

1. The language of the manuscript still needs to be polished to improve accuracy throughout the whole manuscript, especially in the results and discussion sections (see my comments in the attached review report).

2. How is the affinity between the mAb and Ov myosin?

3. Does the mAb recognize the host myosin? I would like to suggest including a homology analysis on the parasite myosin and host orthologs in the manuscript, especially focusing on the binding sites predicted by the molecular modeling.

4. The discussion section can still be deepened. Can authors explain why the mAb cannot recognize the trimmed myosin (spanning nt. 120-2,256 encoding 712 aa) in the ELISA test (Fig 2B).

**Reviewer #2: ** This manuscript primarily focuses on identifying the target antigen in Opisthorchis viverrini recognized by the monoclonal antibody (clone KKU505) developed in previous research. However, there are still several significant areas where additional experiments and further clarification are needed.

1. 1. Can the KKU505 monoclonal antibody detect the ES products of O. viverrini? If it can, the authors should conduct target identification of KKU505 within the ES products of O. viverrini.

2. 2. The Western blot shows a broad area of reactivity. Why did the authors select only two distinct bands for analysis? Why not include the entire reactive region?

3. Since 1D gel-mass spectrometry may lack accuracy, the authors should use additional methods in parallel, such as immunoprecipitation or 2DE-immunomics.

4. What is the specific epitope recognized by the KKU505 monoclonal antibody?

5. For the LC-MS/MS results, please provide a complete list of peptides identified through mass spectrometry, in a supplementary table to ensure unbiased selection.

6. Ideally, this study should include identification of the targets specifically recognized by the KKU505 monoclonal antibody in feces and urine.

6. PLOS authors have the option to publish the peer review history of their article (what does this mean? ). If published, this will include your full peer review and any attached files.

**Do you want your identity to be public for this peer review?** For information about this choice, including consent withdrawal, please see our Privacy Policy .

Reviewer #1: **Yes: ** Pengfei Cai

Reviewer #2: No

---

## [Author Response · Author response to Decision Letter 1]

4 Mar 2025

We appreciate the valuable feedback provided by the reviewers and the editorial team. We have carefully addressed all comments from the editor and reviewers, as detailed in our “Response to Reviewers” document. The western blot results in the manuscript are represented as full membrane. We believe that these revisions have enhanced the quality of our manuscript and improved its clarity and scientific rigor.

---

## [Decision Letter · Decision Letter 1]

22 Apr 2025

Identification and characterization of a target antigen recognized by the monoclonal antibody against Opisthorchis viverrini

PONE-D-24-36944R1

Dear Dr. Thanan,

We’re pleased to inform you that your manuscript has been judged scientifically suitable for publication and will be formally accepted for publication once it meets all outstanding technical requirements.

Kind regards,

Marcello Otake Sato, Ph.D., D.V.M.

Academic Editor

PLOS ONE

Additional Editor Comments (optional):

Reviewers' comments:

Reviewer's Responses to Questions

**Comments to the Author**

1. If the authors have adequately addressed your comments raised in a previous round of review and you feel that this manuscript is now acceptable for publication, you may indicate that here to bypass the “Comments to the Author” section, enter your conflict of interest statement in the “Confidential to Editor” section, and submit your "Accept" recommendation.

Reviewer #1: (No Response)

Reviewer #2: All comments have been addressed

2. Is the manuscript technically sound, and do the data support the conclusions?

Reviewer #1: Yes

Reviewer #2: Yes

3. Has the statistical analysis been performed appropriately and rigorously? 

Reviewer #1: Yes

Reviewer #2: Yes

4. Have the authors made all data underlying the findings in their manuscript fully available?

Reviewer #1: Yes

Reviewer #2: Yes

5. Is the manuscript presented in an intelligible fashion and written in standard English?

Reviewer #1: Yes

Reviewer #2: Yes

6. Review Comments to the Author

Reviewer #1: Thank you for addressing our comments in the revised version. I am satisfied that all my concerns have been resolved.

Some minor points:

Ln 130-133, these two sentences are repetitive.

Ln 341-6, the paragraph is repetitive to the description in the methods. Better to delete it.

Ln 347, PCR with T7 promotor primers (make sure this is correct)

Ln 348-9, the sentence is unclear

Reviewer #2: This manuscript can be accepted. The revised version addressed all of the concerning point with ethics, well analysis and well written.

7. PLOS authors have the option to publish the peer review history of their article (what does this mean? ). If published, this will include your full peer review and any attached files.

**Do you want your identity to be public for this peer review?** For information about this choice, including consent withdrawal, please see our Privacy Policy .

Reviewer #1: No

Reviewer #2: No

---

## [Editor Report · Acceptance letter]

PONE-D-24-36944R1

PLOS ONE

Dear Dr. Thanan,

I'm pleased to inform you that your manuscript has been deemed suitable for publication in PLOS ONE. Congratulations! Your manuscript is now being handed over to our production team.

Kind regards,

on behalf of

Dr. Marcello Otake Sato

Academic Editor

PLOS ONE